# Experimental Investigation of Instabilities on Different Scales in Compressive Fatigue Testing of Composites

Andreas Baumann *[ID] and Joachim Hausmann

Department Fatigue and Life Time Predicition, Leibniz-Institut für Verbundwerkstoffe GmbH, 67663 Kaiserslautern, Germany; joachim.hausmann@ivw.uni-kl.de
* Correspondence: andreas.baumann@ivw.uni-kl.de; Tel.: +49-631-2017-320

**Abstract:** Compression testing of continuous fiber reinforced materials is challenging, because a great number of competing failure modes and instabilities on different length scales have to be considered. In comparison to tensile testing, the results are more affected by the chosen test set-up. Effects introduced by the test set-up as well as the type of damage in continuous fiber reinforced materials are mainly investigated for quasi-static loading. This is not the case for cyclic compression loading. Neither standardized methods nor a great deal of literature for reference exists. The aim of this work is to increase the understanding by analyzing the potential effects the set-up in fatigue loading might have on the damage for two common testing strategies by fatigue tests, load increase creep tests and supplementary analytical models. The results show that damage modes can be altered by the testing strategy for the investigated woven glass fiber reinforced polyamide 6. The tools both experimentally and analytically provide the basis to choose the correct set-up in future investigations.

**Keywords:** fatigue; compression-compression; thermoplastic; glass fiber fabric; anti-buckling support; buckling; composite

## 1. Introduction

The benefits of fiber-reinforced materials are well known and proven for a lot of applications like car bodies, airframes and maritime applications. High specific stiffness and corrosion resistance are only some of the features, which make these materials a potential choice for product developers. However, in many cases fatigue loading can be challenging in the design process. Beside the fact that fatigue testing is especially time consuming and requires specific testing capabilities, it is also a topic seldom addressed by standards or guidelines. However, a great number of investigations showed that tensile loading in fiber direction is an unproblematic loading condition for continuous reinforced composites under fatigue both for thermoset as well as thermoplastic composites. Some examples can be found in [1–3], where moderate stiffness degradation is an indicator for limited fatigue damage. The literature on cyclic compression tests is small in comparison [4–10]. Both factors along with a better material performance lead to the paradigm to use designs where the material is loaded in tension, only. Numerous studies on quasi-static compression properties help to overcome this paradigm for quasi-static loading. Furthermore, standardized methods and great efforts to improve those methods make the characterization comparable over many different studies. This is not the case for compressive fatigue testing. A transfer of procedures and methods from quasi-static testing assumes similar damage mechanisms for both loading cases, which is unlikely considering the results in [10,11].

This leads inevitably to the tasks of qualifying testing procedures for compressive fatigue testing of composites and strongly related to an improved understanding of the dominating damage mechanisms. Compressive in this sense subsumes especially those loading ratios R (ratio of min. to max. stress in one load cycle), which includes no tensile stress within a load cycle (R > 1). It is the aim of this research to understand the effects of the testing method on the compression-compression fatigue behavior.

### 1.1. Problems Associated with Compressive Fatigue Testing

The critical load, which marks the point of instability, is one of the major challenges faced in compression testing of composites. This challenge arises because the majority of composites used are plate like and have a small thickness compared to all other dimensions. As a result, most specimens are slender. Two main strategies to cope with this challenge can be found. First, small gage lengths are used, which lead to a higher buckling load relative to the material's failure load [7]. Second, to use longer specimens and to increase the buckling load by the use of a supporting structure, so-called anti-buckling device [4]. However, both methods have their drawbacks. For example, could the anti-buckling device act as a second load path beside the specimen or damage propagation could be slowed. On the other hand, stress inhomogeneity at the load introduction is less important compared to specimens with a short free length between the fixtures. Furthermore, this inhomogeneity sets a lower limit for the gage length [12]. Load introduction at the specimen ends is one measure to reduce this effect [13]. In addition to load introduction, fixture alignment is of great importance. For many quasi-static compression set-ups, perfect alignment of the specimen ends is crucial and suspected to effect the property scattering [14]. To verify the alignment of the testing machine or the specimen a transducer with strain gages can be used ASTM E 1012 [15]. Despite this guidance on machine alignment, the problem remains for most cases, as tabs are necessary. Especially for a short gage length as only slight tapering of the specimen is possible the specimen's ends must be protected and the stress reduced by tabs. It is hard to accomplish perfectly symmetric tab bonding. Consequently, a bending surveillance is established for quasi-static compression testing by two strain gages on opposite surfaces of the specimen [16]. The bending surveillance is difficult to accomplish for fatigue testing [17] as strain gages and their attachment also have a limited fatigue life. As a result, changes in sensitivity and zero drift can occur [18]. Even though non-contact strain measuring gets increasingly better, a documentation of bending over the experimental fatigue cycles is rarely done. Compared to a greater gage length the influence of a slight misalignment on the testing results is more severe [19].

It can be concluded that global buckling is avoided by both methods but potentially different results might be obtained. How much the testing method effects the results could be influenced by the occurring damage mode, load introduction and generally size effects.

### 1.2. Damage Modes under Compression-Compression Fatigue Loading

Typical damage modes observed for composites under quasi-static compressive loading are fiber-kinking, fiber crushing, splitting, delamination and shear band formation [20]. However, only delaminations have been reported to be a precursor for final fatigue failure [11] under compressive fatigue loading. As shown by Matondang et al. [10] the use of an anti-buckling guide can also influence the damage induced in the specimen. In their investigation, they found that delaminations starting from the specimen's edges could be delayed, due to constrained edges in thickness direction. These findings are supported by investigations of Kardomateas and Malik [21], who found that mode mixity in the post-buckled state is responsible for the delamination growth. Under the assumption that out of plane movement is constrained for supported specimens, the ratio of opening (Mode I) and shearing stresses (Mode II) at the tip of the delamination is restricted to Mode II and therefore slower delamination growth can be expected [22]. On the other hand, failure size with respect to the gage length can also effect the results. Delaminations starting from the specimen edges are described as localized, in contrast to typically larger delaminations as a result of e.g., impact damage. Nevertheless, it seems unlikely that the delamination size is negligible for small gage lengths, despite the lack of reported work on this topic. Investigations on the initiation of delaminations are also sparse, but as Bak et al. [23] pointed out it is generally thought that delamination initiation is a result of small material defects. For a given number of samples used to determine the S-N curve and a fixed distribution of flaws, the overall test volume for short and long gage length specimens and thus the number of flaws differ. This in turn can affect the likelihood to pick a flawed specimen. This size

effect has already been investigated for quasi-static loading by several authors [24–26], but no common conclusion has been reached. Anyway, the role of size effects could be more important for fatigue loading as damage progression may be influenced. Kardomateas [22] found, for example, that delamination propagation is relevant for cyclic but not for static loading. Another failure mode, which might be influenced by the choice of gage length is fiber kinking. Even though a size effect is reported by Bažant et al. [24] for this damage mode, it is primarily concerned with the specimen's width and not the gage length. Kink bands propagate typically along the width of the specimen inclined to the specimen's axis. This incline can lead to a lower bound for the gage length. As Vogler and Kyriakides [27] pointed out the initiation phase can be influenced by the length of the specimen. On the other hand, their research showed also that by laterally constraining the specimen the formation of kink bands could be influenced. For a laterally constrained specimen surface, the kink band propagates in width direction whereas for unconstrained specimens the free edges are especially prone for out-of-plane kinking described as barreling. Ueda et al. [28] found also a combination of in-plane and out of plane kinking by in situ CT observation. However, fiber kinking was initiated by fiber failure near the specimen's edge. These findings suggest that the use of an anti-buckling guide might not necessarily affect the initiation of a kink band but it is likely that the propagation is influenced.

From the studied literature, it becomes clear that the two most common testing methods might lead to different failure modes or an altered damage propagation. Delamination and fiber kinking seem to be especially prone to be influenced by an anti-buckling guide or the measuring length. Test set-ups using an intermediate layer of crushable material [17] or just a layer of PTFE [29] between the anti-buckling plates gave rise to a number of questions. First, it is unclear how much a laterally supported specimen comes into contact with the anti-buckling guides, secondly, how much the supports alter the damage progression and finally what is the overall response in terms of fatigue life under compression-compression loading. To the authors' knowledge, this research is the first to address those questions and to compare different supporting materials and compression-compression testing strategies directly by characterizing the same composite material in different set-ups. An increased understanding of those effects can benefit future investigations of the compression dominated loading regime of composites. Furthermore, it should help to choose the testing set-up with the best representation of the conditions the material encounters in the final component.

## 2. Materials and Methods

### 2.1. Materials

To investigate the effect the testing strategy has on the compressive fatigue damage behavior an organo-sheet material Tepex® dynalite 102-RG600(x)/47% supplied by Lanxess Bond Laminates [30] is used. The material is a twill weave glass fiber fabric with a PA6 matrix and an overall fiber volume content of 47% consisting of four layers fabric resulting in a thickness of 2 mm. Composites with woven reinforcement seem especially prone to the initiation of kink bands or localized delaminations, due to the microscopic structure. It is expected that the undulations are natural initiations points. For the calculation of kink band propagation stress the material supplier provided the pure resin properties listed in Table 1. The matrix hardening exponent in the Ramberg–Osggood material model and the minimum tangent modulus are chosen according to Skovsgaard and Jensen [31]. The minimum tangent modulus for the matrix material is a lower bound for the matrix materials stiffness, which restricts the Ramberg–Osgood relation for high strains [31].

**Table 1.** Material properties Tepex dynalite and the constituents.

| Property | | | Constituent | Source |
|---|---|---|---|---|
| elastic modulus | 1000 | MPa | matrix | [32] |
| minimum elastic modulus | 10 | MPa | matrix | [31] |
| yield stress | 40 | MPa | matrix | [32] |
| hardening exponent | 4 | | matrix | [31] |
| Poisson's ratio | 0.39 | | | [3] |
| fiber volume content | 0.6 | | roving | [3] |
| elastic modulus | 19.6 | GPa | laminate | [3] |

The long gage length set-up with its anti-buckling device and the additional intermediate materials is shown in Figure 1. The tapered specimen (dimensions in Figure 1) is placed between two layers of aramid, cardboard or PTFE material, subsequently referred to as intermediate material (see also Supplementary Material S1). The tapering allows for testing without tabs. The intermediate layers were two aramid honeycomb materials (thickness 15 and 20 mm), cardboard honeycombs (thickness 30 mm) and PTFE film. Characterization of the materials used for the intermediate layer was part of this research in order to use a homogeneous database.

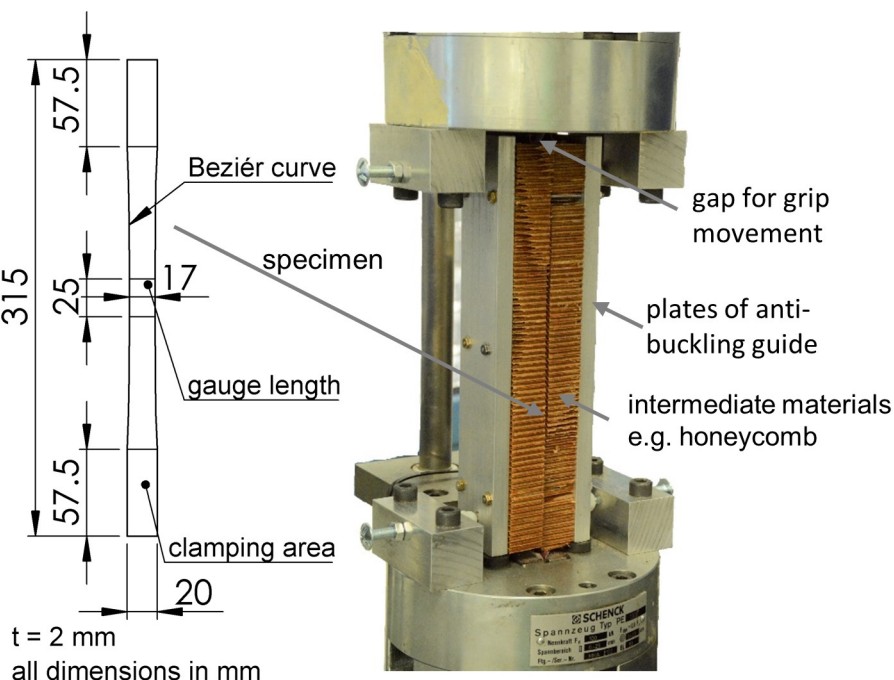

**Figure 1.** Specimen geometry and intermediate material in the test set-up.

### 2.2. Experiments

Figure 2 shows an overview of all testing set-ups used in this investigation. Cyclic compression-compression tests are performed with two different gage lengths. Loading cycles for these tests are characterized by a loading ratio of R = 10 and a frequency of 1 Hz. Pure shear loading as load introduction method is used for both gage lengths. Special care was taken to assure perfectly aligned clamps, verified by a metal specimen fitted with strain gages. Table 2 shows the compression-compression fatigue tests with the according load levels applied. In brackets is the number of samples planned for the test. Instead of using three specimens on four load levels, the fatigue tests focused on the load level of −180 MPa, because of the large scattering in resulting lifetime. Similarly, the testing results with 10 mm free length increased the number of tested samples.

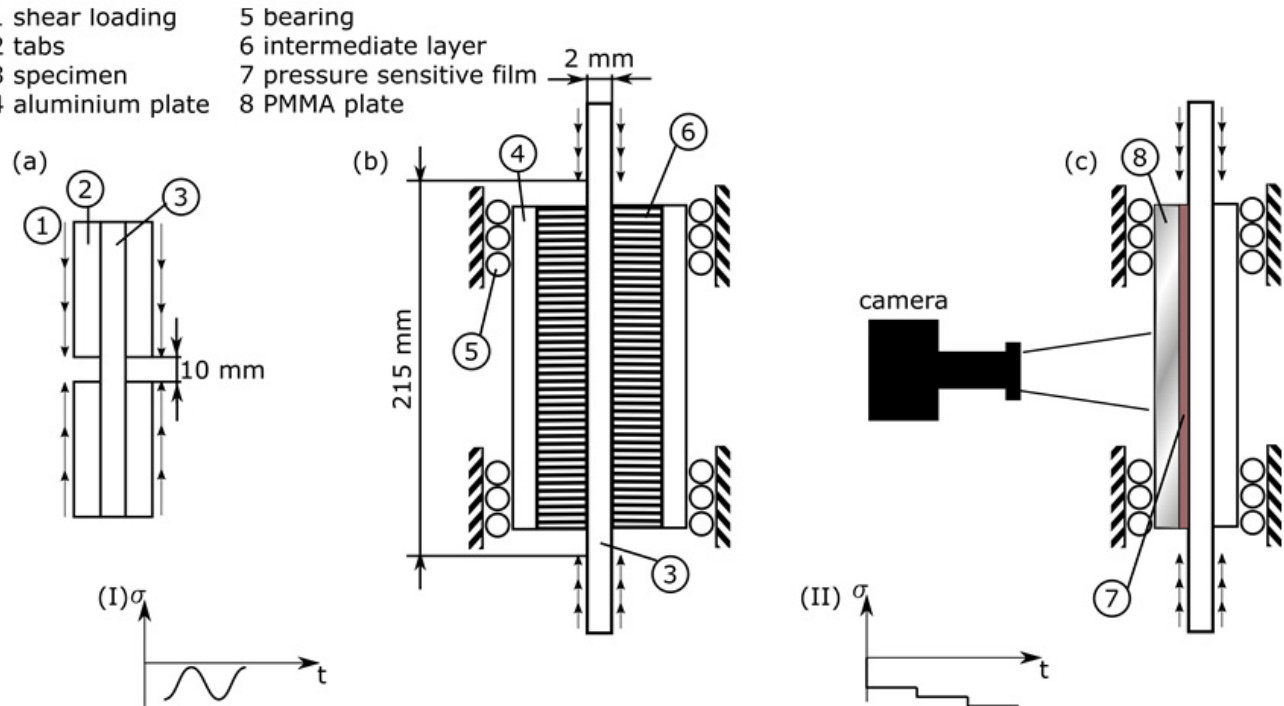

**Figure 2.** Experimental set-ups and corresponding test procedure; (**a-I**) short gage length cyclic, (**b-I**) long supported gage length cyclic, (**b-II**) long supported gage length in LIC-test, (**c-II**) LIC-test with pressure sensitive film.

**Table 2.** Compression-compression fatigue tests.

| Gage Length | 10 mm | | 215 mm | |
|---|---|---|---|---|
| | **Load Levels** | **No of Samples** | **Load Levels** | **No of Samples** |
| | | | −145 MPa | 1 (3) * |
| | −180 MPa | 10 (5) * | −160 MPa | 2 (3) * |
| | | | −180 MPa | 7 (3) * |
| | | | −205 MPa | 2 (3) * |
| **Intermediate layers** | - | | honeycomb 20 mm | |
| **Testing frequency** | 1 Hz | | | |
| **Loading ratio R** | 10 | | | |

* Tested number of samples (planned number of samples).

Load increase creep (LIC) test on the same servo-hydraulic testing rig complement the long gage length fatigue tests. Recent experiments have shown, that these tests lead to similar damage modes compared to the fatigue tests but with less scattering in terms of time and therefore significantly less testing time. Furthermore, this method allows for the comparison of different intermediate layer materials in the supporting set-up with respect to the composites damage modes and the ability to prevent global buckling for different load levels. Additional LIC tests are performed with a pressure sensitive film (Fuji Film Prescale Low LW 2.5–10 MPa) for a better understanding of the damage initiation and shape and size of contact zones to the anti-buckling device. The pressure build up against the supporting structure in these tests is documented by in-situ observation of the coloring of the pressure film through a PMMA plate. The coloring and therefore the pressure is only qualitatively studied, as the calibration curves of the pressure sensitive films are only valid under a defined loading sequence, humidity and illumination wavelength.

In all LIC tests the first load level is −90 MPa, which was then increased every 10 h by −10 MPa. The first load level is derived from the mean stress of the corresponding

S-N curve, which results in similar loading times. A mean stress of −90 MPa leads to approximately 36k cycles. This corresponds to a loading time of about 10 h with 1 Hz. This approach is based on the observed strong fatigue-creep interaction of the laminate under compressive fatigue loading. Under the assumption that the creep failure is also the main cause of failure under compressive fatigue loading, the chosen time increment is the minimum time to cause failure for load levels below −90 MPa. All tests concerning the anti-buckling device (Figure 2b,c) make use of the tapered tensile specimen (TTS) [33].

For a better interpretation of the different results in the LIC test, the intermediate layers/materials are also characterized. First, in quasi-static compression tests, and second, in indentation tests with different roller diameters. The quasi-static compression test is performed with a crosshead speed of v = 10%/min on eight samples 50 × 50 × t for the aramid honeycomb materials and 80 × 80 × t for the cardboard material. A larger cross-section of the cardboard material should overcome the coarser material structure (e.g., cell size). The set-up and procedure is based on DIN EN ISO 844. The effect of cell size and material structure is separately evaluated by indentation tests with a cylindrical indenter. To estimate the sensitivity to localized contact different indenter diameters (10, 15, 20 mm) are used. The indentation experiments help to evaluate the effect of the supporting structure (cell size, homogeneity) and its contact with the specimen. All indentation experiments used a maximum indentation depth of 5 mm and a testing speed of 2 mm/min. In order to get an estimate for the elastic properties of the PTFE tape material additional compression tests were performed with small piston diameter 10 mm and a reduced speed of only 0.5 mm/min. A piston instead of compression plates as well as the reduced speed were chosen in order protect the measuring equipment and to minimize alignment errors.

### 2.3. Analytical Models

Specimen and anti-buckling guide might interact in three different ways. For one, the intended global interaction will take place, which makes the unbuckled state energetically stable. For geometrically perfect set-ups and ideally stiff supporting plates, only axial compression of the specimen is possible. However, in practical set-ups these conditions are not possible. Small clearances necessary to avoid friction and elastic supports make the problem of global interaction effectively one of constrained or elastically supported buckling. The second and third interaction are usually unintended and are localized, namely propagation of kink bands and delamination growth. All three interactions are triggered by an instability, on either a macro (global buckling), meso (delamination) or micro (fiber kinking) scale. Recognizing this, it becomes clear why it is hard to prohibit one but not all of the aforementioned instabilities. In particular, initiation and growth of kink bands and delamination are considered as inherent material behavior and should evolve equally in lab scale tests and components. In addition to the following analytical methods, it is also possible to use numerical algorithms to model the effects local instabilities might have on the component as well as the lab-scale specimen. One such method could be the arc-length method, which is especially suitable to model the effects after bifurcation [31,34,35].

#### 2.3.1. Microscopic Instability

Besides fiber kinking the single fiber buckling can also be seen as a microscopic instability. However, as it is unlikely that the anti-buckling guide affects single fiber buckling this instability is neglected at this stage. The numerical investigation of Diaz Montiel and Venkataraman [35] for cyclic compression-compression loading show that after an initial bifurcation of the fiber bundle at a high stress the second and following loading cycles show a much lower stress to reach the same strain state. Matrix plasticization and damage of the fiber–matrix interface are seen as main reasons that the initially small misalignment grows and decreases the kink stress. This propagation stress can be seen as the necessary stress to allow for steady kink band growth under cyclic loading. Kink band propagation after bifurcation can be calculated by an analytical model first formulated by Christoffersen and Henrik [36] and later used by Skovsgaard and Jensen [31]. They

assumed a uniaxial stress state meaning that only stresses in fiber direction are present. The use of an anti-buckling guide can lead to through thickness stresses either by pre-loading of the plates or because the thickness of the specimen increases, due to kink-band formation. A free body diagram taken far from the clamps is as shown in Figure 3. Taking the same notation as [31] but extending the model by through thickness stresses the resulting equilibrium formulas are as below. Regarding the symbols, please refer to Figure 3.

$$-\sigma_{11}^o \cos 2\beta - \sigma_{22}^o(\phi)\sin^2\beta - \sigma_{11}^i \cos^2\chi + \sigma_{22}^i \sin^2\chi - 2\sigma_{12}^i \sin\chi\cos\chi = 0 \tag{1}$$

$$\sigma_{11}^o \cos\beta\sin\beta - \sigma_{22}^o(\phi)\sin\beta\cos\beta + \sigma_{11}^i \cos\chi\sin\chi + \sigma_{22}^i \sin\chi\cos\chi - \sigma_{12}^i\left(\cos^2\chi - \sin^2\chi\right) = 0 \tag{2}$$

$$\chi = \beta - \phi \tag{3}$$

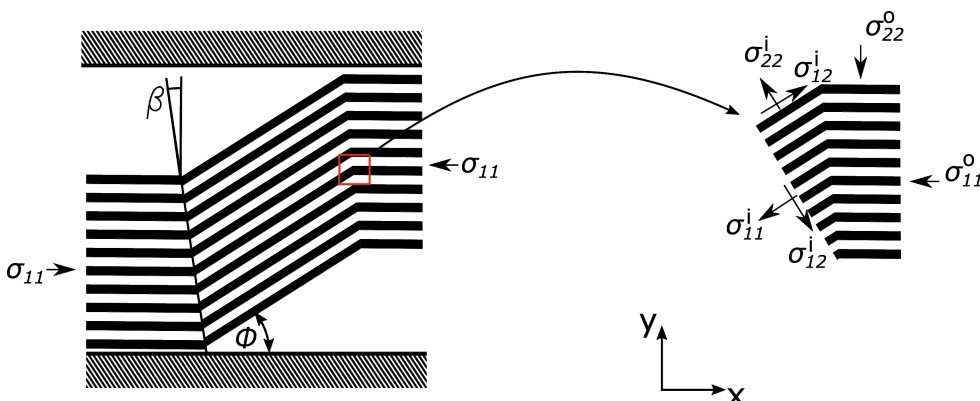

**Figure 3.** Free body diagram of a kink band forming under constrained conditions.

Solving for $\sigma_{11}^o$ allows to express the external work by internal stresses under the assumption of inextensible fibers. This along with the assumption that for the lock-up state internal and external work are in equilibrium leads to the lock-up angle and steady state kink band propagation angle. For more details on the solution and the assumed material model, readers are referred to Skovsgaard and Jensen's paper [31]. A normalized length of one is taken into account for the calculation of the external work.

The through thickness stress $\sigma_{22}^o$ can either be taken to be constant or as a function of $\Phi$. By doing so, the modified external work becomes (4). It is interesting to note that, due to the sign convention the work done by the compressive stress $\sigma_{11}^o$ leads to an increase of the kink bands energy whereupon at the same time the kink band transfers energy in thickness direction by expanding against the resisting supports.

$$W^E = -\sigma_{11}^o(1 - \cos\phi) + \sigma_{22}^o \sin\phi \tag{4}$$

To calculate the necessary propagation stress $\sigma_{11}^o$ for different initial misalignment angles $\beta$ and supporting stresses $\sigma_{22}^o$ the crossing point for internal and external work as a function of $\Phi$ must be calculated [31]. Kink band propagation in the supported cases is evaluated by either specifying a constant value for $\sigma_{22}^o$ or by defining a stiffness for the supporting material $E_s$. This leads to Equation (5), where $L$ is the kink band length.

$$\sigma_{22}^o = E_s * L \sin\phi \tag{5}$$

### 2.3.2. Macroscopic Stability Failure

Delaminations develop in two stages, first initiation and second growth in the buckled state of the delaminated layer. Most models on delamination growth assume a small, delaminated area, which is already stressed beyond the critical load and are therefore concerned with the growth phase [22]. It is most likely that by supporting the specimen

initiation and growth of delaminations can be altered. Perfectly supported specimens are in contact with the supporting material from the beginning and thereby the critical load for the delaminated area can be calculated for the elastically supported column or plate [19]. As shown in the section 'global buckling', an elastic support can increase the bifurcation load and buckling order. However, as not much is known on the length scales and mechanisms in the initiation phase, growth in a non-perfect supported case is seen as more relevant and the corresponding models are reported below. For this case, the literature on constrained buckling provides some insights. Available solutions incorporate analytical solutions for small deformation by buckling between rigid walls and frictionless contact [37]. For higher order buckling and large deformations, the elliptical integrals of the elastica must be used [37,38]. Especially the latter case leads to complex formulations. Even though some specific solutions for special cases with springy walls exist [39], only the rigid frictionless case is considered here.

The walls with which the delaminated section interacts, are the intermediate material of the anti-buckling device and the base material of the specimen. Because the initial stiffness (see Figure 10 in the results Section 3.2.1) of all supporting materials is relatively low, and furthermore, the surface of the organo sheet is not perfectly plane an initial, a clearing like configuration is assumed for the delaminated section. Friction is also disregarded, as the models should only provide some insight on how the anti-buckling guides can influence the delamination growth. Under these assumptions, Chai [37] provides a sequential solution for the two dimensional case. Four phases for the delaminated section can be distinguished (i) pre-buckling, (ii) post-buckling with no contact, (iii) post-buckling with point contact and finally, (iv) post-buckling with line contact. For higher loads, additional bifurcations arise. For the first buckling phase (ii) a cosine approximation can be used to describe the shape of the delaminated section. However, as the load remains constant only displacement can be used as controlling parameter. In this buckling phase, no reactional force R exists. Delamination growth is nevertheless possible due to the reactional moment and the in-plane force submitted to the buckled section. In Phase (ii), the reactional moment M is directly correlated to the buckled shape. In Phase (iii), when point contact and a reactional force exist the reactional moment can be calculated by (6). The factor $\frac{Ph}{2}$ gives the maximum reactional moment for the current axial force $P$ and slit width $h$. The current bending moment is normalized by this factor in order to compare the maximum bending moment for the constrained case to the unconstrained elastica. Figure 4 compares the constrained case in different buckling phases to the unconstrained case. The free body diagram is taken at the inflection point for each of the Phases (ii) to (iv). This leads to the expression below.

$$M = \frac{Ph}{2} - \frac{RL}{8} \qquad (6)$$

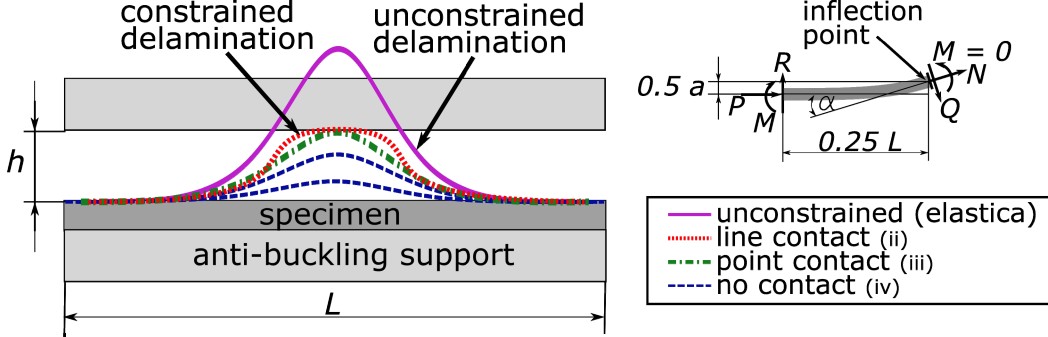

**Figure 4.** Delamination in the constrained and unconstrained case and corresponding free body diagram.

For details on the formulation of the linear approximation, the reader is referred to Chai's paper [37]. In addition to this linear small deformation model, the elastica

for the clamped-clamped configuration of Timoshenko and Gere [19] is used. This large deformation theory is necessary for the unconstrained case and can be used to test the linear approximation for the observed localized buckles and their dimensions. By normalization, it is possible to free the comparison of geometry and material parameters (namely elastic modulus *E* and moment of inertia *I*). This results in Equation (8). In all calculations and figures only the first buckling mode is used and therefore the number of half waves is *n* = 1. *K* is the complete elliptical integral of the first kind. Equation (9) gives the amplitude of the elastica in the unconstrained case.

$$P = \frac{EI}{(0.25L)^2} K(\sin(\frac{\alpha}{2})) \tag{7}$$

$$\frac{P}{P_{crit}(n = 1)} = \frac{4}{\pi^2} K(\sin(\frac{\alpha}{2})) \tag{8}$$

$$a = 4(\sin(\frac{\alpha}{2})) \sqrt{\frac{EI}{P}} \tag{9}$$

2.3.3. Global Buckling

The anti-buckling support should increase the global buckling load to loads beyond the applied cyclic loads. Timoshenko and Gere [19] provide a solution to calculate the critical load for the elastically supported column. $E_s$ is the elastic modulus of the supporting material, which can be converted by the specimens width *b* divided by the intermediate materials thickness *t* into the definition of Timoshenko and Gere's modulus of the foundation. The critical buckling load is the minimum found by for different buckling orders *n*.

$$P_{crit.} = Min(P) = \frac{\pi 2 EI}{L2} \left( n^2 + \frac{\frac{b}{t} E_s L^4}{n2 \pi^4 EI} \right) \text{ for } n \in \mathbb{N} \tag{10}$$

## 3. Results
### 3.1. Modelling Results
3.1.1. Microscopic Stability Failure

In a first approximation, the kink band model of Section 2.3.1 is used under the assumption of constant supporting (through thickness) stress. The subscripts for the stress notation is accordingly. Two effects are considered, first, different initial misalignment angels of the fiber bundles and, second, different supporting stresses. Figure 5 shows the effect of both variables on the steady state kink band propagation stress. As already shown for the unsupported case by other authors [31,35], an increased initial misalignment angel reduces the stress necessary to initiate and propagate a kink band. A reduced propagation stress can also be found for the supported case. With increasing supporting stress, the propagation stress also increases. In terms of a supporting the anti-buckling guide, it might be that a very stiff supporting material or highly pre-tightened plates can increase the propagation stress. Except for the PTFE tape, it is likely that the anti-buckling guides investigated here do not affect kink band propagation. This is because usually a full surface contact does not exist between the intermediate material and the specimen, either due to the nature of the intermediate material (cells of the honeycomb material) or because of uneven surfaces of the specimen and the support. Furthermore, the stiffness of the supporting material is low with respect to the expected deformations, as the out of plane deformation of the kink-band is only in the length scale of a few micrometers [40].

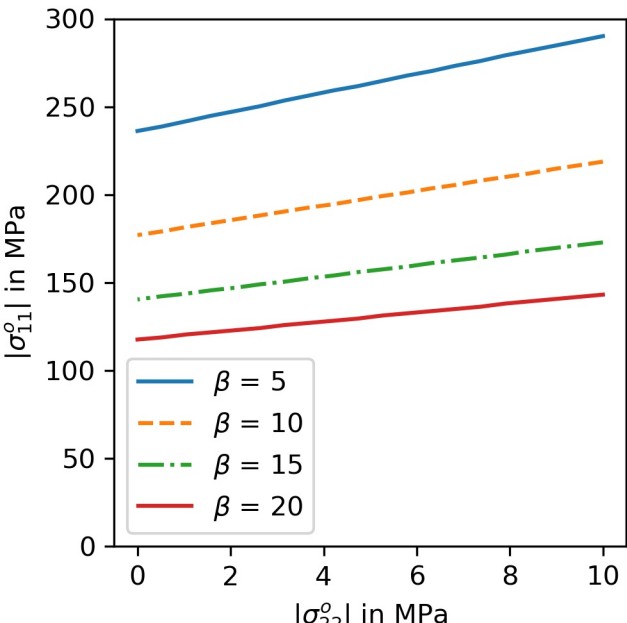

**Figure 5.** Kink band steady state propagation stress for different misalignment angles and supporting stresses.

However, for large misalignment angels of β = 20° and equally high lock-up angles of 43° a kink band of the length L = 200 μm [40] could theoretically be effected. In order to reach a supporting stress of 5 MPa a stiffness of 37 MPa/mm is necessary. Figure 6 shows the lock-up angle as a function of initial misalignment in the unsupported case and the supported case. The elastically supported case yields lock-up angles between those two lines. It is interesting to note that with decreasing initial misalignment angle β the effect is bigger. In terms of fiber orientation, this means that highly aligned unidirectional materials could potentially be more affected.

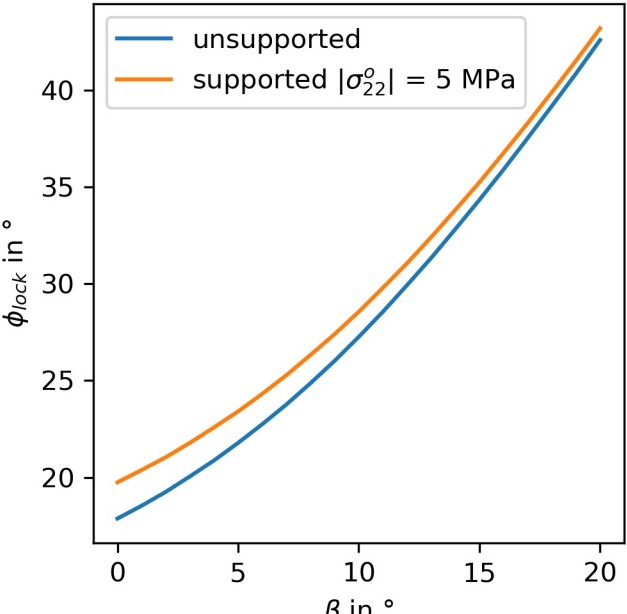

**Figure 6.** Lock-up angle and initial misalignment angle for the supported and unsupported case.

### 3.1.2. Macroscopic Stability Failure

In order to estimate the effect the supporting structure has on the delamination growth of a composite under compressive loading the models for frictionless walls are used. To get a hint if the linear approximation reported by Chai is applicable the observed delamination height of 0.075 mm is taken as an estimate of the clearance between the constraining surfaces. The unbuckled length of the delaminated section is taken as approximation with 1 mm. In the linear cosine approximation, the normalized axial load is taken to be unity until contact with the walls occurs. The normalized load to cause an amplitude of 0.075 mm is 1.015 calculated by Equation (9) for the amplitude of the elastica. Therefore, the linear approximation underestimates the load slightly by 1.5% and equally the reactional moment. Figure 7 shows the reactional moment at the ends of the delaminated strip. In Phase (i), no reactional moment exists. In the first post-buckling Phase (ii) the bending moment increases until contact with constraining intermediate material (support) occurs. It is interesting to observe that the normalized bending moment steadily increases in the unconstrained case, whereas it starts to decrease in Phase (iii) of the post-buckling phase of the constrained case. In terms of stress intensity factors and mode mixity, it is obvious that these parameters are drastically different in the constrained and unconstrained case. However, it must be noted that the increase in reactional moment is most likely limited by the materials strength. Maximum values of five in Figure 7 for the normalized reactional moment correspond to extreme rotations of 90° at the inflection point. In comparison, the constrained case submits much higher axial forces without excessive bending of the delaminated strip. Figure 8 shows the transition to higher order buckling. With the assumed clearance of the constraining surfaces (0.075 mm), the next bifurcation point is reached only for loads 16 times the initial buckling load of the delaminated strip. Higher order buckling of the delaminated area is therefore only relevant for a greater clearance or delaminations of greater length. From Figure 8, it can also be seen that the reactional force depends on the applied axial force P as well as the current buckling order. For cyclic loading, this would suggest repeated wall contact and furthermore repeated loading of the intermediate material.

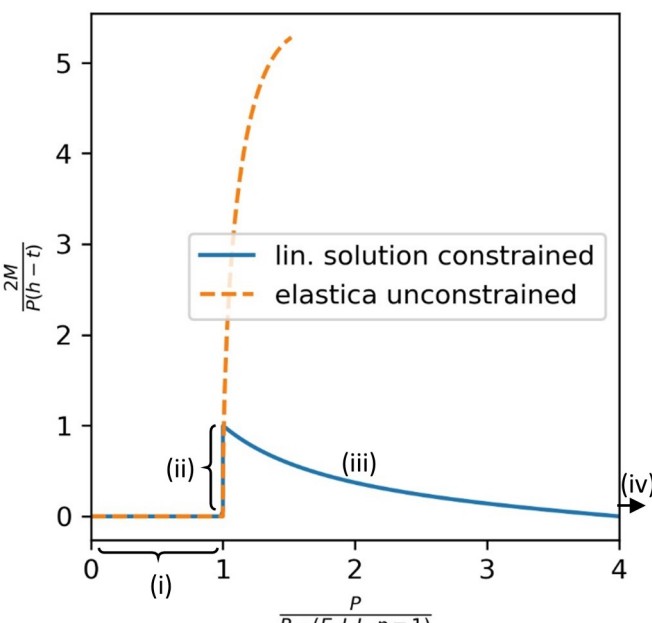

**Figure 7.** Normalized axial load and bending moment in the constrained and unconstrained case.

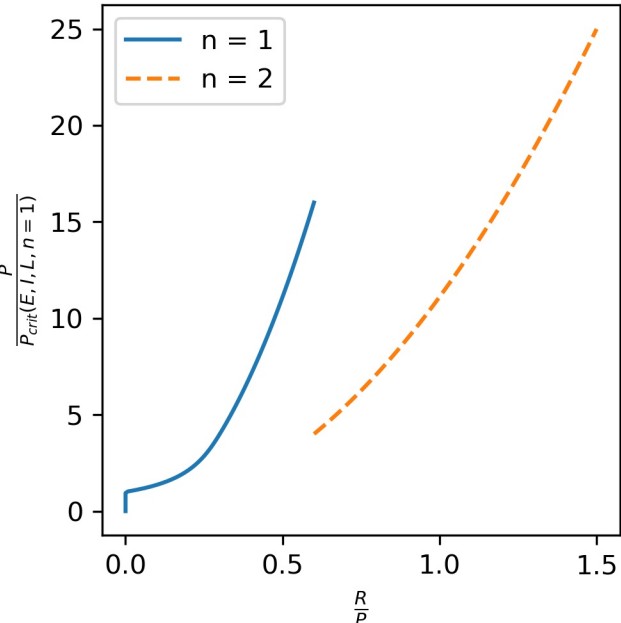

**Figure 8.** Linearized model in the constrained case, reactional force and bifurcation.

### 3.1.3. Global Stability Failure

An elastically supported specimen with perfect contact to the intermediate material shows two dependencies with respect to the supporting stiffness. First, with increasing stiffness of the intermediate material, the lowest buckling load is shifted to higher buckling modes ($n$ is increased). Second, the critical buckling load also increases relative to the unsupported case. Figure 9 shows the global buckling load for different buckling orders ($n$). The buckling loads are calculated with the measured stiffness of the intermediate materials and a length of 200 mm in-between the anti-buckling guide. A hinged case is assumed, because of an unconstrained region (approx. 7.5 mm) above and below the anti-buckling guide. The value of Table 1 for the elastic modulus of the composite material is used.

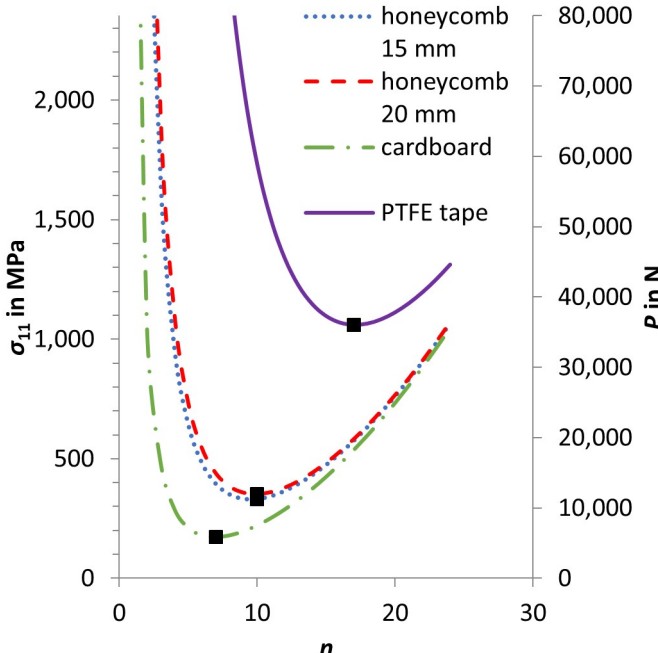

**Figure 9.** Global buckling load as a function of buckling order for the investigated materials.

### 3.2. Experimental Results

3.2.1. Intermediate Material Characterization

The first step of the experimental program is the intermediate material characterization by quasi-static compression tests. The corresponding stress-displacement curves are reported in Figure 10. Initially the stress-displacement curve shows a progressive behavior, which is the result of the compression plates coming increasingly into contact with samples. It is interesting to note that the stress-displacement behavior is steady until a critical crushing stress is reached. The buckling of the cell walls mainly causes this instability. This feature is most pronounced for the aramid honeycombs and not relevant for the PTFE film due to its monolithic structure.

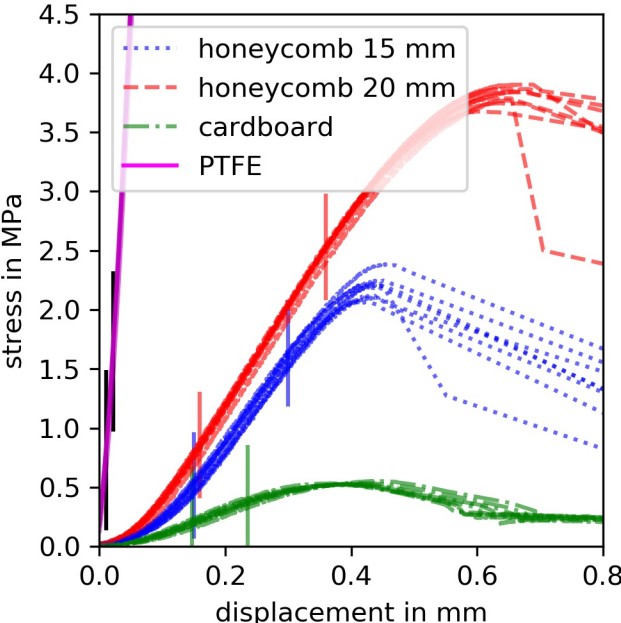

**Figure 10.** Stress-displacement response of the intermediate materials; vertical lines mark the section for the linear stiffness evaluation.

Besides the conventional elastic modulus, a stiffness is evaluated, to account for different thicknesses of the intermediate materials. This measure is defined as elastic modulus divided by thickness, which results in the unit $N/mm^3$. In Figure 10, the vertical lines represent start and end for the linear stiffness and elastic modulus evaluation. The initial stiffness was evaluated for all materials from zero to 0.2% strain. Table 3 summarizes all materials used as intermediate layers and their respective properties. The stiffness of the PTFE material is by far greater compared to the rest of materials under consideration. The elastic modulus in thickness direction should be taken as an estimate because despite machine stiffness compensation of the measured displacement the results can be more affected by other factors compared to the stiffness.

**Table 3.** Properties of the intermediate materials.

| Designation | Elastic Modulus MPa | | Stiffness N/mm³ | Crushing Stress MPa | Cell Diameter mm | Cell Wall Thickness mm | Density kg/m³ | t mm |
|---|---|---|---|---|---|---|---|---|
| | Initial | Linear | Linear | | | | | |
| honeycomb 15 mm | 12.8 | 109 | 7.40 | 2.19 | 3.6 | 0.03 | 50.8 | 14.7 |
| honeycomb 20 mm | 34.7 | 168 | 8.48 | 3.80 | 3.2 | 0.05 | 62.3 | 19.8 |
| cardboard | 17.6 | 60.8 | 2.06 | 0.53 | 10.0 | 0.13 | 51.7 | 29.5 |
| PTFE tape | - | ~8.42 | 76.5 | - | - | - | - | 0.11 |

The indentation tests show a similar displacement-force behavior compared to the compression tests (Figure 11). The initial linear segment is followed by a sudden decrease in stiffness with the onset of crushing. As before, the crushing is most pronounced for the stiffer aramid honeycombs. In order to evaluate the effect of different indenter diameters on the crushing force the momentary stiffness and its first drop of 50% is used as a measure for the onset of crushing Figure 12. In order to generate a gradient signal a Butterworth low pass filter with cut-off frequency 1 Hz is used as signal conditioning tool. It is interesting to note that the crushing force of the honeycomb 15 mm material is in the same order of the cardboard material. However, the stiffness is much greater for both honeycomb materials Figure 13. Furthermore, it is interesting to note that the stiffness increase as a function of indenter diameter is different for all three materials. In terms of damage size, this would suggest different supporting stiffness for different failure sizes. However, crushing force and stiffness show similar tendencies.

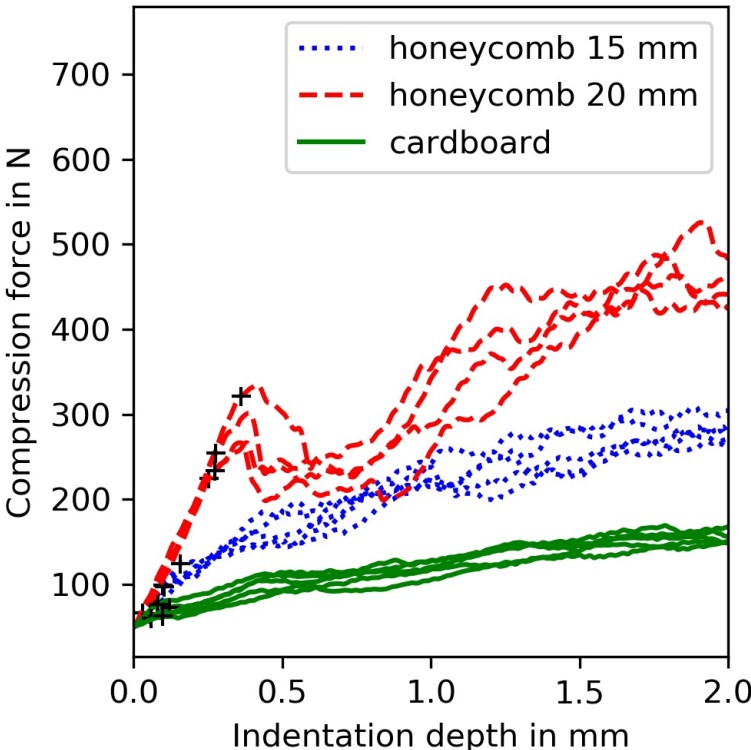

**Figure 11.** Indentation experiments with indenter diameter 15 mm.

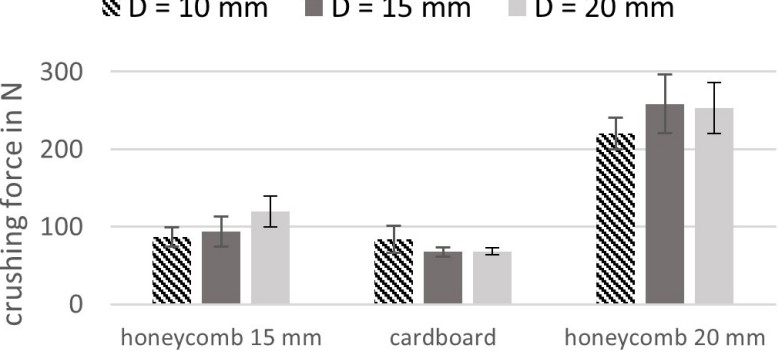

**Figure 12.** Crushing force for different indenter diameters.

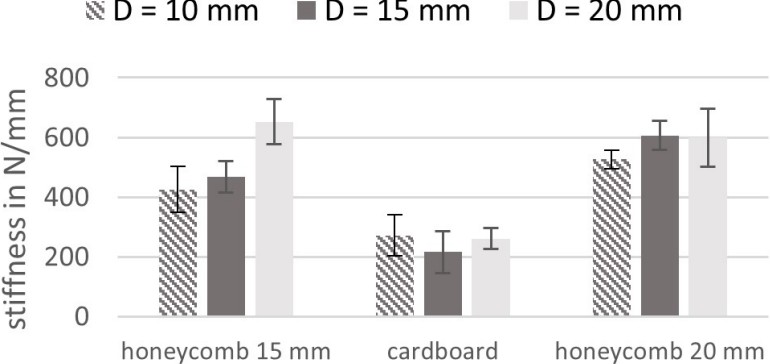

**Figure 13.** Stiffness for different indenter diameters.

### 3.2.2. Cyclic Loading

Figure 14 shows the compression-compression fatigue curve of the organosheet material evaluated with the aramid honeycomb 20 mm intermediate supporting material. This material was chosen for the fatigue test because promising results have been obtained for alternating loading in the same test set-up (R = −74 [17]). All specimens failed inside the gage length. The indentation of the honeycomb materials clearly show that global buckling did not occur in any of the fatigue tests. The failure type could be described as through thickness shear according to DIN EN ISO 14126. With a Wöhler exponent of 36.7 the S-N curve is comparatively flat (8.33 for R = 0.1 [3]). However, scattering is large. The dashed lines show the 95% prognosis interval, which assumes a logarithmic normal distribution. The basis for the calculation of the prognosis interval is the standard error of the prediction for the whole set of data points, which is then used to estimate the prognosis error of the regression. Besides, in this research, we used software-implemented procedures for calculation, details on the calculation can also be found in [41]. Scattering becomes especially obvious by comparing different stress prognoses for, e.g., $10^6$ cycles. If failure should be avoided the calculated regression stress must be reduced by 34 MPa to expect only 5% of the specimens to fail. This is a reduction of the allowable stress by 22%. Severe scattering is especially visible for the load level of 180 MPa, which some of the specimens endured for up to $5 \times 10^6$ cycles without failure, whereas failure could also occur after several hundred cycles. One possible explanation for the observed scattering could be the type of pre-failure damage. As summarized in Section 3.2.5, small material buckles form on the specimen's surface, especially for the lower loaded specimens. However, this phenomenon is not evenly distributed but can be found only once or twice on one sample. In some of the specimens, the localized buckles had different amplitudes. This gave the impression that several possible failure locations were active but in different stages to final failure. The even load introduction would allow only differences in materials properties or mesoscopic geometry to be the cause for this potential pre-failure stage. In a first working hypothesis, it is assumed that matrix creep leads to this localized buckling, which is initiated by sparsely scattered geometrical inhomogeneity. This observation and hypothesis lead to further investigation of the material in LIC tests and observation of the surface deformation.

Unfortunately, the single load level comparison between short gage length specimens and large gage length specimens was not possible due to debonding of the tabs. Despite additional effort with different tabbing materials, primers and bonding agents debonding remained a problem. However, for some specimens localized buckling was also visible despite the problems associated with this testing strategy. The localized buckle endured for a great number of cycles. However, additional tests are necessary to be conclusive.

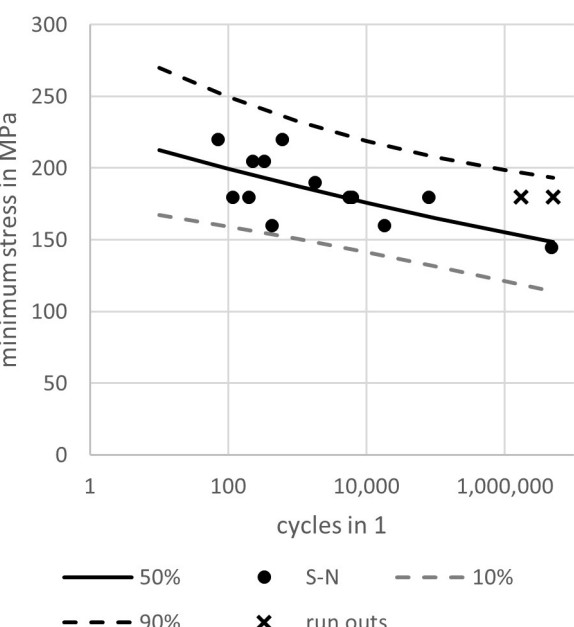

**Figure 14.** S-N curve with aramid honeycomb 20 mm support of the specimens.

### 3.2.3. Load Increase Creep (LIC) Tests

The LIC tests showed considerable effect of the supporting material. Intermediate layers, which resulted in global buckling of the specimen, were visible by damage marks or imprints on the supporting material. This was obvious for the cardboard supported specimens. As predicted by the models for elastically supported specimens the global buckling order increased. However, not from one to seven half waves but from first order to third order for the cardboard material. Furthermore, the critical buckling load/stress was over predicted with 174 MPa instead of the achieved 128 MPa. Therefore, the failure mode with cardboard as intermediate material is unacceptable and excluded in the subsequent discussion.

The hypothesis that matrix creep is not negligible is further underlined by the time to failure after increasing the load (Figure 15). If only linear elastic stability failure on different scales would be activated, this would result in failure instantly after or while reaching the next load level. However, the mean time to failure after reaching the next load level is 3.4 h. The distribution of times to failure show a clear tendency to shorter times, but also that the majority of the specimens took more than 2 h to fail after increasing the load. Regarding the different supporting materials, no distinct differences can be seen with respect to this parameter. The specimens show similar pre-failure damage compared to the compression-compression fatigue samples.

Differences between the supporting materials become evident with respect to the failure load. The highest stresses were sustained by specimens supported by aluminum plates with PTFE sheets (198 MPa), followed by the 15 mm and 20 mm honeycombs (168 and 160 MPa). Cardboard supported specimens lead to premature failure, due to the global instability (128 MPa). Considering the fact that the initial stiffness (see Table 3) of the cardboard material is higher compared to 15 and 20 mm honeycombs, it seems that this parameter has no obvious effect on the ability of the supporting plate to prevent global buckling. Cell size or other parameters seem to be more relevant. The effect a second load path might have was also checked. This additional verification was done by cutting a specimen in the middle and pulling the two halves apart, in the supported configuration.

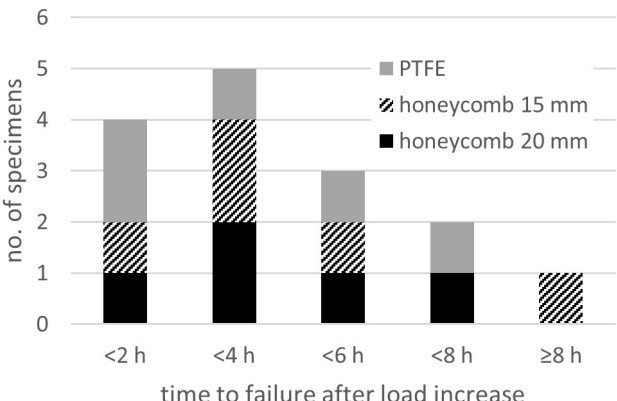

**Figure 15.** Time to failure after load increase.

### 3.2.4. Load Increase Test with Pressure Sensitive Film

In preliminary tests, the necessary pressure range for the film was found to be optimal between 2.5–10 MPa. Furthermore, the pressure sensitive film gave the opportunity to verify the amount and distribution of contact pressure in the chosen assembly process. The preliminary tests with more sensitive pressure films (LLW 0.5–2.5 MPa [42]) sometimes showed contact pressure by the tightening of the assembly screws in the lower part of this pressure range. The loading sequence is identical to the LIC tests with different intermediate layers. Figure 16 shows the last minutes before final failure of one exemplary specimen. All six specimens failed finally from the edge. However, often this final failure is initiated by a localized high-pressure zone, which develops suddenly within the capture period of 3 min between frames. It is interesting to note that this localized deformation initiates additional deformation in neighboring areas. These areas are typically at the edge of the specimen and lead to final failure.

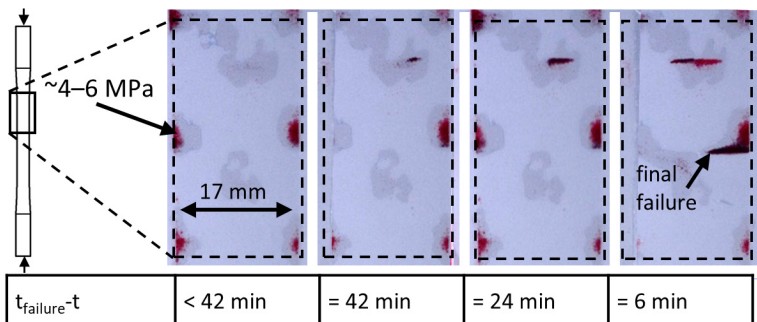

**Figure 16.** Failure evolution on load level 160 MPa with LW pressure sensitive film (red indicates pressure acting on the anti-buckling device).

### 3.2.5. Compressive Damage Investigations

Similar damage modes could be observed in the form of localized buckling over all tests. Figure 17 shows two exemplary specimens. A white light interferometer scan of the same area of the LIC specimen is shown in Figure 18. The calculations in Section 2.3.2 make use of the measured peak height of 0.075 mm. No buckles were visible on the specimen surface before the respective loading.

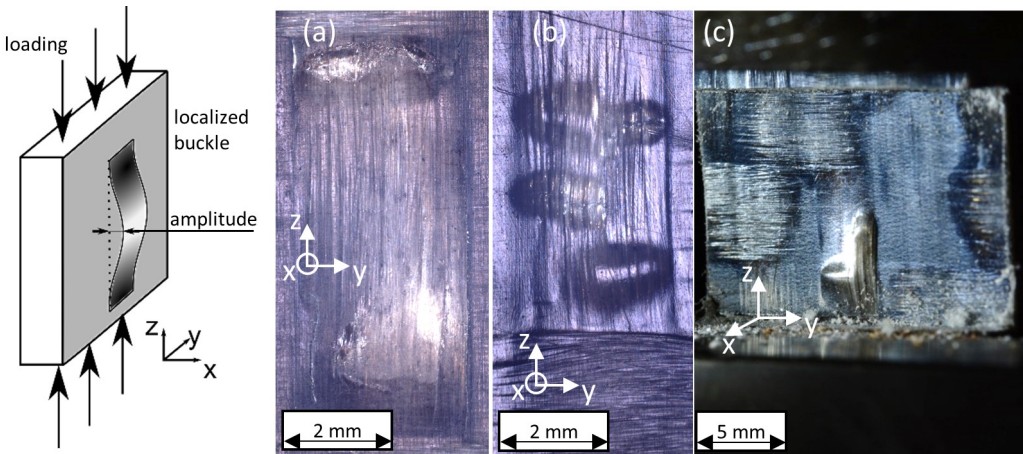

**Figure 17.** Localized buckling with confocal bright field illumination: (**a**) fatigue specimen with long gage length; (**b**) LIC test specimen; (**c**) in-situ observation short gage length.

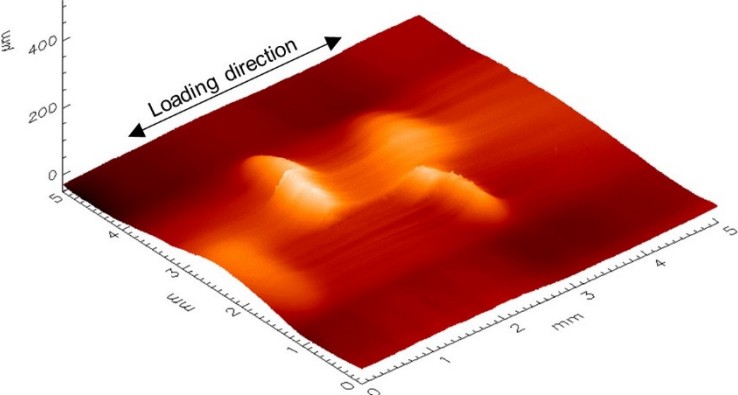

**Figure 18.** White light interferometric diagram of the LIC specimen.

## 4. Discussion

### 4.1. Effect of the Supporting Material

In contrast to the initial stiffness, a possible correlation between the stiffness of the supporting material and the failure type is observed. Instead of comparing the elastic modulus of the supporting materials, it is better to use stiffness (e.g., elastic modulus divided by thickness) as a measure for comparison, because the shape and reactional forces in a slightly buckled column depend on the resulting deformation of the intermediate material and not its strain. A stiffness of $2\,\text{N/mm}^3$ or below is not enough to prevent global buckling in the testing configuration despite the predictions for an elastically supported column. One possible explanation for the over predicted buckling load could be that the results in Figure 9 do not incorporate the shear compliance of the laminate. Other likely reasons could be not ideally straight specimens in combination with a low crushing stress of the cardboard honeycombs. The failure load of the LIC test can be correlated with the stiffness of the intermediate material especially for the high differences—Figure 19. The influence of the crushing stress on the failure load level in the LIC can only be evaluated by comparing Figures 19 and 20. The highest load achieved in the LIC test is for both aramid honeycomb materials similar. In terms of crushing stress, a difference would be expected whereas in terms of stiffness not. Furthermore, no crushing is possible for the PTFE film. These observations make the stiffness of the intermediate material in the linear-elastic region of the stress-strain curve the main factor influencing the failure load.

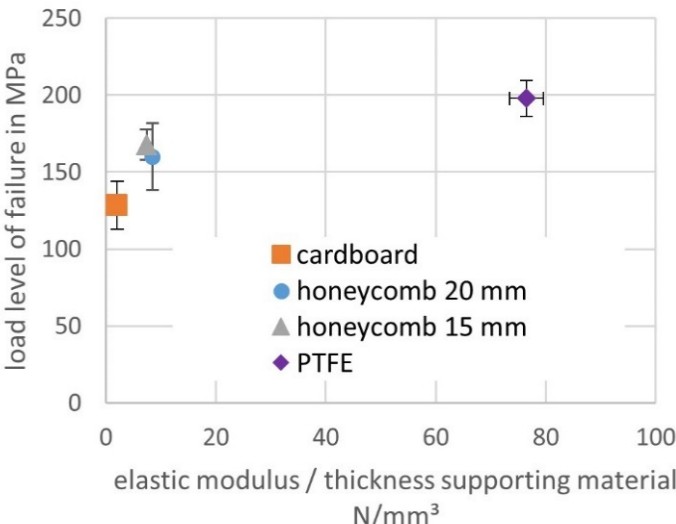

**Figure 19.** Elastic modulus of the supporting materials vs. load level of failure in the LIC test.

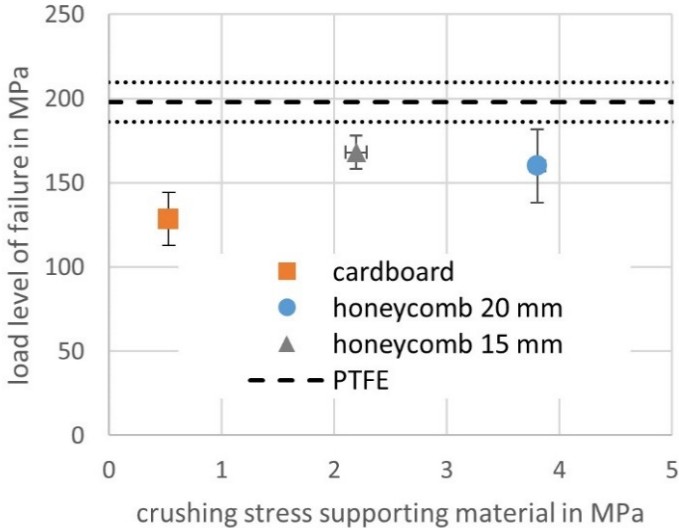

**Figure 20.** Crushing stress of the supporting materials vs. load level of failure in the LIC test.

### 4.2. Testing Strategy

From the premature failure of the cardboard supported specimens, it can be concluded that the analytical model for elastically supported columns can be used to define a minimum required supporting stiffness. However, a slight over prediction must be expected. The analytical models predict for the aramid honeycomb materials a global buckling load slightly above the fatigue testing loads and in none of the tests with those intermediate materials global buckling was observed. However, from the experiments with stiffer intermediate materials and calculated results for fiber kinking and delamination growth it must be noted that the supporting structure likely enhances the failure load of those damage modes. Therefore, the investigated aramid honeycomb materials should have the least effect on the testing results, as the critical load for global buckling is only 50% above the highest load level used for fatigue testing. Besides the intermediate material's stiffness, the crushing stress is also an important factor. From the LIC test with a pressure sensitive film, it was obvious that the pressure necessary to keep the specimen in a straight configuration is in the same order as the crushing stress of some of the materials. This adds another statistical feature, which increases the complexity of correct choice for the compressive fatigue testing set-up.

On the other hand, considering the experiences with a short unsupported specimen, it becomes evident that tabbing introduces another factor, which can prolong testing times by unacceptable failure. Even worse, it can lead to global buckling, which should be avoided by design and may stay unnoticed. In addition to these problems, macroscopic damage development namely localized delamination might also be effected by the constraining effect of the grips. This becomes evident through the comparison of damage development in the LIC test with pressure sensitive films and the observations in the fatigue tests with a gage length of 10 mm. The LIC test showed that localized delamination would introduce damage and failure in neighboring areas, which lay inside the gripped zone of the short gage length specimen. This sequence of events is less likely as the neighboring area is inside the grips of a short gage length specimen.

### 4.3. Compressive Fatigue Damage

The techniques used to compare different testing strategies also led to further insights into the compressive behavior of the material under investigation. The compressive fatigue testing results suggested a large effect of local material inhomogeneity on the durability of the specimens under these loading conditions. The localized buckling or delamination near the specimen's surface, which could be observed in fatigue testing with and without support and also in the LIC test suggest a more pronounced role of matrix creep in addition to a material inhomogeneity. These observations were further underlined by the LIC test with additional pressure sensitive film, which showed a steady delamination growth, which started several minutes before final failure. Furthermore, it was evident that these localized buckling redistributed the stresses. By this redistribution, other areas typically at the specimen edge were overloaded and the specimen collapsed. To this point, it remains to clarify if the material inhomogeneity is the type of a geometric inhomogeneity or just small defects in the form of pores or debonding. However, large areas of the material were already investigated by micro sectioning and μCT and no pores or debonding was evident [3]. This would suggest the geometric inhomogeneity near the specimen's surface to be the main initiator of failure. Incorporating this assumption large scattering of the fatigue testing results might in part be explained by the distribution of geometric inhomogeneity near the specimen surface. However, a correlation between scattering in the compressive fatigue testing results would need a complementary geometry analysis of a statistically significant part of the organo sheet material. Independent of what defect initiates the localized buckles, a sequence of events can be reckoned under the assumption that the LIC and the fatigue tests show a common failure propagation. First a local material separation occurs, which allows a fraction of the fibers of a roving (oriented along the loading direction) to buckle. This small buckle grows in length and amplitude both because of mean stress (creep) and loading amplitude (fatigue). This local defect redistributes some of the load to neighboring areas, and furthermore, introduces asymmetry in the stress distribution in the adjacent cross sections. Finally, this inhomogeneity in stress leads to the final specimen failure. The final fracture surface is inclined approximately 30–60° to the specimens axis.

The comparison of the damage observed in this study to earlier studies with different loading ratios (R = 0.1 [3] and R = −0.74 [17]) but the same specimen geometry and material show a very distinct effect the loading ratio has on the type of damage. For moderate compressive loads as in the alternative loading R = −0.74 no localized buckles were reported [17]. Therefore, the sequence of events discerned from the LIC test and the compression-compression fatigue test show, that there could be a critical compressive mean stress to activate these damage mechanisms. As a result of different damage mechanisms from tension dominated loading to compression dominated loading, the asymmetry of the constant lifetime diagrams is not only the result of different failure stresses but also of different slopes of the S-N curve.

## 5. Conclusions

From the results conclusions on the testing strategy and the material's behavior can be drawn.

Material behavior:

- The loading ratio shows a great effect on the damage mechanisms of the Tepex® dynalite material in comparison to the current, and to earlier published results both in the resulting S-N curve and the acting damage mechanisms
- Localized buckling can be identified as a pre-failure mechanism for compression-compression loading and is precursor for final failure. This could also be used as warning sign for the inspection of parts in-service.
- Fatigue and creep are equally important for compression-compression loading regimes. A critical mean stress and loading time might activate the damage mechanism and explain some of the scattering observed.

Testing strategy:

- Testing results obtained with supported specimens are different for a high stiffness of the supporting material, which could be shown both experimentally and by analytical models.
- High contact pressures applied by the anti-buckling device are likely affecting the propagation stress of kink bands and must equally be considered besides the stiffness of the supporting material.
- The elastically supported column model used in this work has proven to be a viable tool to estimate the minimum stiffness for the intermediate layers of the anti-buckling device.
- Small gage length testing with tabbed specimens might also affect the testing results because only a small material volume is tested and local asymmetric stress distributions do not lead to final failure in contrast to the materials behavior in later components.

From this, it can be concluded that the testing strategy must be fitted to the expected damage mode. A better understanding of differences and similarities between component failure and lab scale failure under compression fatigue is needed to overcome the prejudice that compressive fatigue must be avoided even though extremely shallow S-N curves can be expected.

**Supplementary Materials:** The following are available online at https://www.mdpi.com/article/10.3390/jcs5040114/s1, Figure S1: intermediate materials after indentation test.

**Author Contributions:** Conceptualization, methodology, investigation; writing—original draft preparation, A.B.; resources, writing—review and editing, supervision, funding acquisition J.H. All authors have read and agreed to the published version of the manuscript.

**Funding:** This research was in part funded by the "Zentrales Innovationsprogramm Mittelstand" (grant number: ZF4052306RR6).

**Institutional Review Board Statement:** Not applicable.

**Informed Consent Statement:** Not applicable.

**Data Availability Statement:** Data is contained within the article.

**Acknowledgments:** The authors acknowledge the material data provided by Bond Laminates.

**Conflicts of Interest:** The authors declare no conflict of interest.

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
