# Peer review of "Experimental Investigation of Instabilities on Different Scales in Compressive Fatigue Testing of Composites"

_jcs, doi:10.3390/jcs5040114_

Round 1
Reviewer 1 Report
The manuscript presented the investigation of the effect of experimental compressive fatigue setup for composites on the resulting damage for two common testing strategies, i.e., load increase creep tests and supplementary analytical models. The manuscript is well written, and the topic falls within the scope of the journal. Before recommending publication, I would like to ask the authors to address the remarks listed below.
(1) In the second line of page 2 where the authors wrote “which includes no tensile stress within a load cycle (R>=1)”, does the authors mean R>=0?
(2) Why did not the authors consider the transverse shear stress components in equation (1)?
(3) The curves in figure 4 are hard to distinguish. The authors are suggested to use different colors to label different curves to improve the reading experience.
(4) The authors are suggested to briefly mention a few recent publications on the numerical modeling algorithms of instabilities: (a) doi.org/10.1016/j.cma.2019.112585, (b) doi.org/10.1061/(ASCE)EM.1943-7889.0001263
Reviewer 2 Report
Thank You for this interesting research. In the reviewers opinion the paper deserves to have a chance to be published after slight changes. The main strength of the paper lies in the well planned and performed experimental procedure. The experimental test resulst are also rare for such loading ratio of R=10 which is also a reason to publish the paper, as such results are of practical use for enginners and researchers. The authors should put more emphasis on the mean stress/strain effects, which are involved in the fatigue process. It plays a significant role in the fatigue lifetime assessment process and it should be more highlighted. More information about the error analysis is required as some of the graphs presenting the scatter of results are based on single values. The conclusions should be more concise and refer to experimental numerical results. In the current form some of the conclusions are rather observations. I suggest to publish the paper after major revision.
Reviewer 3 Report
Comments
This paper studied fatigue testing of composites. The outcome is interesting for readers. However, there are several aspects that need to be improved. The reviewer can only recommend for publication if the author satisfactorily address the following comments in the revised version.
- The author need to provide stress ratio, frequency and applied load range for fatigue test in a Table.
- How many samples were tested at each load level?
- Why compression-compression fatigue test was chosen over tension-tension fatigue test?
- Table 1 need reformatting. The left most column is not easily readable.
- Spelling check: associatet
- The failure mechanism of the specimen should be discussed more clearly
- The novelty of the study should be highlighted clearly at the end of introduction section. How this study is different from the published study in literature?
- How the outcome of this study will benefit researchers and end users? This need to be highlighted in introduction or end of conclusion.
- The background study on the fatigue behaviour of composites should be improved. Recently fatigue behaviour of composite laminates were investigated for epoxy based [Ref: Testing and modelling the fatigue behaviour of GFRP composites–Effect of stress level, stress concentration and frequency] and polyester and vinyl ester based [Ref: Tensile Fatigue Behavior of Polyester and Vinyl Ester Based GFRP Laminates—A Comparative Evaluation] laminates. Suggest to include them in introduction section with proper citations to improve the background study.
I would be happy to see the revised version to understand how these comments are being addressed.
Round 2
Reviewer 3 Report
I have no further comments.